# GUIDED EXPLORATION WITH PROXIMAL POLICY OPTIMIZATION USING A SINGLE DEMONSTRATION

## ABSTRACT

Solving sparse reward tasks through exploration is one of the major challenges in deep reinforcement learning, especially in three-dimensional, partially-observable environments. Critically, the algorithm proposed in this article uses a single human demonstration to solve hard-exploration problems. We train an agent on a combination of demonstrations and own experience to solve problems with variable initial conditions. We adapt this idea and integrate it with the proximal policy optimization (PPO). The agent is able to increase its performance and to tackle harder problems by replaying its own past trajectories prioritizing them based on the obtained reward and the maximum value of the trajectory. We compare variations of this algorithm to different imitation learning algorithms on a set of hard-exploration tasks in the Animal-AI Olympics environment. To the best of our knowledge, learning a task in a three-dimensional environment with comparable difficulty has never been considered before using only one human demonstration.

## 1 INTRODUCTION

Exploration is one of the most challenging problems in reinforcement learning. Although significant progress has been made in solving this problem in recent years (Badia et al., 2020a;b; Burda et al., 2018; Pathak et al., 2017; Ecoffet et al., 2019), many of the solutions rely on very strong assumptions such as access to the agent position and deterministic, fully observable or low dimensional environments. For three-dimensional, partially observable stochastic environments with no access to the agent position the exploration problem still remains unsolved.

Learning from demonstrations allows to directly bypass this problem but it only works under specific conditions, e.g. large number of demonstration trajectories or access to an expert to query for optimal actions (Ross et al., 2011). Furthermore, a policy learned from demonstrations in this way will only be optimal in the vicinity of the demonstrator trajectories and only for the initial conditions of such trajectories. Our work is at the intersection of these two classes of solutions, exploration and imitation, in that we use only one trajectory from the demonstrator per problem to solve hard-exploration tasks [1].

This approach has been explored before by Paine et al. (2019) (for a thorough comparison see Section 5.2). We propose the first implementation based on on-policy algorithms, in particular PPO. Henceforth we refer to the version of the algorithm we put forward as PPO + Demonstrations (PPO+D). Our contributions can be summarized as follows:

1. In our approach, we treat the demonstrations trajectories as if they were actions taken by the agent in the real environment. We can do this because in PPO the policy only specifies a distribution over the action space. We force the actions of the policy to equal the demonstration actions instead of sampling from the policy distribution and in this way we accelerate the exploration phase. We use importance sampling to account for sampling from a distribution different than the policy. The frequency with which the demonstrations are sampled depends on an adjustable hyperparameter $\rho$, as described in Paine et al. (2019).

2. Our algorithm includes the successful trajectories in the replay buffer during training and treats them as demonstrations.

---

[1]A video showing the experimental results is available at `https://www.youtube.com/playlist?list=PLBeSdcnDP2WFQWLBrLGSkwtitneOelcm-`

3. The non-successful trajectories are ranked according to their maximum estimated value and are stored in a different replay buffer.

4. We mitigate the effect of catastrophic forgetting by using the maximum reward and the maximum estimated value of the trajectories to prioritize experience replay.

PPO+D is only in part on-policy as a fraction of its experience comes from a replay buffer and therefore was collected by an older version of the policy. The importance sampling is limited to the action loss in PPO and does not adjust the target value in the value loss as in Espeholt et al. (2018).

We found that this new algorithm is capable of solving problems that are not solvable using normal PPO, behavioral cloning, GAIL, nor combining behavioral cloning and PPO. PPO+D is very easy to implement by only slightly modifying the PPO algorithm. Crucially, the learned policy is significantly different and more efficient than the demonstration used in the training.

To test this new algorithm we created a benchmark of hard-exploration problems of varying levels of difficulty using the Animal-AI Olympics challenge environment (Beyret et al., 2019; Crosby et al., 2019). All the tasks considered have random initial position and the PPO policy uses entropy regularization so that memorizing the sequence of actions of the demonstration will not suffice to complete any of the tasks.

## 2 RELATED WORK

Different attempts have been made to use demonstrations efficiently in hard-exploration problems. In Salimans & Chen (2018) the agent is able to learn a robust policy only using one demonstration. The demonstration is replayed for $n$ steps after which the agent is left to learn on its own. By incrementally decreasing the number of steps $n$, the agent learns a policy that is robust to randomness (introduced in this case by using sticky actions or no-ops (Machado et al., 2018), since the game is fundamentally a deterministic one). However, this approach only works in a fully deterministic environment since replaying the actions has the role of resetting the environment to a particular configuration. This method of resetting is obviously not feasible in a stochastic environment.

Ecoffet et al. (2019) is another algorithm that largely exploits the determinism of the environment by resetting to previously reached states. It works by maximizing the diversity of the states reached. It is able to identify such diversity among states by down-sampling the observations, and by considering as different states only those observations that have a different down-sampled image. This works remarkably well in two dimensional environments, but is unlikely to work on three-dimensional, stochastic environments.

Self-supervised prediction approaches, such as Pathak et al. (2017); Burda et al. (2018); Schmidhuber (2010); Badia et al. (2020b) have been used successfully in stochastic environments, although they are less effective in three-dimensional environments. Another class of algorithms designed to solve exploration problems are count-based methods (Tang et al., 2017; Bellemare et al., 2016; Ostrovski et al., 2017). These algorithms keep track of the states where the agent has been before (if we are dealing with a prohibitive number of states, the dimensions along which the agent moves can be discretized), and give the agent an incentive (in the form of a bonus reward) for visiting new states. This approach assumes we have a reliable way to track the position of the agent.

An empirical comparison between these two classes of exploration algorithms was made in Baker et al. (2019), where agents compete against each other leveraging the use of tools that they learn to manipulate. They found the count-based approach works better when applied not only to the agent position, but also to relevant entities in the environment (such as the positions of objects). When only given access to the agent position, the RND algorithm (Burda et al., 2018) was found to lead to a higher performance.

Some other works focus on leveraging the use of expert demonstrations while still maximizing the reward. These methods allow the agent to outperform the expert demonstration in many problems. Hester et al. (2018) combines temporal difference updates in the Deep Q-Network (DQN) algorithm with supervised classification of the demonstrator's actions. Kang et al. (2018) proposes to effectively leverage available demonstrations to guide exploration through enforcing occupancy measure matching between the learned policy and current demonstrations.

Other approaches, such as Duan et al. (2017); Zhou et al. (2019) pursue a meta-learning strategy where the agent learns to learn from demonstrations, such approaches are perhaps the most promising, but they require at least a demonstration for each task for a subset of all tasks.

Generative adversarial imitation learning (GAIL) (Ho & Ermon, 2016) has never been successfully applied to complex partially observable environments that require memory (Paine et al., 2019). InfoGAIL (Li et al., 2017) has been used to learn from images but, unlike in this work the policy does not require recurrency to complete the tasks. Behavioral Cloning (BC) is the most basic imitation learning technique, it is equivalent to supervised classification of the demonstrator's actions (Rahmatizadeh et al., 2018). Both GAIL and BC have been used in the Obstacle Tower Challenge (Juliani et al., 2019), but are alone insufficient for solving hard-exploration problems (Nichol, 2019).

Perhaps the article that has the most similarity with this work is Oh et al. (2018). The authors present an off-policy algorithm that learns to reproduce the agent's past good decisions. Their work focuses mainly on advantage actor-critic, but the idea was tested also with PPO. Instead, PPO+D utilizes importance sampling, leverages expert (human) demonstrations, uses prioritized experience replay based on the maximum value of each trajectory to overcome catastrophic forgetting and the ratio of demonstrations replayed during training can be explicitly controlled.

## 3   DEMONSTRATION GUIDED EXPLORATION WITH PPO+D

Our approach attempts to exploit the idea of combining demonstrations with the agent's own experience in an on-policy algorithm such as PPO. This approach is particularly effective for hard-exploration problems. One can view the replay of demonstrations as a possible trajectory of the agent in the current environment. This means that the only point we are interfering with the classical PPO is when sampling, which is substituted by simply replaying the demonstrations. A crucial difference between PPO+D and R2D3 (Paine et al., 2019) is that we do consider sequences that contain entire episodes in them, and therefore using recurrency is much more straightforward. From the perspective of the agent it is as if it is always lucky when sampling the actions, and in doing so it is skipping the exploration phase. The importance sampling formula provides the average rewards over policy $\pi_{\theta'}$ given trajectories generated from a different policy $\pi_\theta(a|s)$:

$$\mathbb{E}_{\pi_{\theta'}}[r_t] = \mathbb{E}_{\pi_\theta}\left[\frac{\pi_{\theta'}(a_t|s_t)}{\pi_\theta(a_t|s_t)}r_t\right], \tag{1}$$

where $r_t = R(r|a_t, s_t)$ is the environment reward given state $s$ and action $a$ at time $t$. $\mathbb{E}_{\pi_\theta}$ indicates that the average is over trajectories drawn from the parameterized policy $\pi_\theta$. The importance sampling term $\frac{\pi_{\theta'}(a_t|s_t)}{\pi_\theta(a_t|s_t)}$ accounts for the correction in the change of the distribution over actions in the policy $\pi_{\theta'}(a_t|s_t)$. By maximizing $\mathbb{E}_{\pi_\theta}\left[\frac{\pi_{\theta'}(a_t|s_t)}{\pi_\theta(a_t|s_t)}r_t\right]$ over the parameters $\theta'$ a new policy is obtained that is on average better than the old one. The PPO loss function is then defined as

$$L_t^{CLIP+VF+S}(\theta) = \mathbb{E}_t\left[L_t^{CLIP}(\theta) + c_1 L_t^{VF}(\theta) + c_2 S^{\pi_\theta}(s_t, a_t)\right], \tag{2}$$

where

$$L_t^{CLIP}(s, a, \theta', \theta) = \min\left(\frac{\pi_{\theta'}(a_t|s_t)}{\pi_\theta(a_t|s_t)} A^{\pi_\theta}(s_t, a_t), \ g(\epsilon, A^{\pi_\theta}(s_t, a_t))\right) \tag{3}$$

and $c_1$ and $c_2$ are coefficients, $L_t^{VF}$ is the squared-error loss $(V_\theta(s_t) - V_t^{targ})^2$, $A^{\pi_\theta}$ is an estimate of the advantage function and $S$ is the entropy of the policy distribution. The entropy term is designed to help keep the search alive by preventing convergence to a single choice of output, especially when several choices all lead to roughly the same reward (Williams & Peng, 1991).

Let $\mathcal{D}$ be the set of trajectories $\tau = (s_0, a_0, s_1, a_1, ...)$ for which we have a demonstration, then $\pi_{\mathcal{D}}(a_t|s_t) = 1$ for $(a_t, s_t)$ in $\mathcal{D}$ and 0 otherwise. This is a policy that only maps observations coming from the demonstration buffer to a distribution over actions. Such distribution assigns probability one to the action taken in the demonstration and zero to all the remaining actions. The algorithm decides where to sample trajectories from each time an episode is completed (for running out of time or for completing the task). We define $\mathcal{D}$ to be the set of all trajectories that can get replayed at any given time $\mathcal{D} = \mathcal{D}_V \cup \mathcal{D}_R$, where $\mathcal{D}_V$ are the trajectories collected prioritizing the value estimate ($V$ stands for value), and $\mathcal{D}_R$ ($R$ stands for reward) contains the initial human demonstration

and successful trajectories the agent collects. The agent samples from the trajectories in $\mathcal{D}_\mathcal{R}$ with probability $\rho$, $\mathcal{D}_\mathcal{V}$ with probability $\phi$ and from the real environment Env with probability $1 - \rho - \phi$ subject to $\rho + \phi \leq 1$.

The behavior of the policy can be defined as follows: $\pi_\theta^{\phi,\rho} = \begin{cases} \pi_{\mathcal{D}_\mathcal{R}}, & \text{if sampled from } \mathcal{D}_\mathcal{R} \\ \pi_{\mathcal{D}_\mathcal{V}}, & \text{if sampled from } \mathcal{D}_\mathcal{V} \\ \pi_\theta, & \text{if sampled from Env} \end{cases}$

In PPO+D we substitute the current policy $\pi_\theta$ with the policy $\pi_\theta^{\phi,\rho}$, since this is the policy used to collect the demonstration trajectories. The clipping term is then changed to correct for it,

$$L_t^{CLIP-PPO+D}(s, a, \theta', \theta) = \min \left( \frac{\pi_{\theta'}(a_t|s_t)}{\pi_\theta^{\phi,\rho}(a_t|s_t)} A^{\pi_\theta^{\phi,\rho}}(s_t, a_t), \ g(\epsilon, A^{\pi_\theta^{\phi,\rho}}(s_t, a_t)) \right). \quad (4)$$

---

**Algorithm 1** PPO+D

---

1: Initialize parameters $\theta$
2: Initialize replay buffer $\mathcal{D}_V \leftarrow \{\}$
3: Initialize replay buffer $\mathcal{D}_R \leftarrow \{\tau_\mathcal{D}\}$
4: Initialize rollout storage $\mathcal{E} \leftarrow \{\}$
5: **for** every update **do**
6:     **for** actors $1, 2, \ldots, N$ **do**
7:         Sample $\tau$ from $\{\mathcal{D}_V, \mathcal{D}_R, \text{Env}\}$
8:         **if** $\tau \in \mathcal{D}_R$ **then**
9:             With probability $\rho$ sample batch of demonstrations
10:             **for** steps $1, 2, \ldots, T$ **do**
11:                 Replay a transition from buffer $s_t, a_t, r_t, s_{t+1} \sim \pi_{\mathcal{D}_R}(a_t|s_t)$
12:                 Store transition $\mathcal{E} \leftarrow \mathcal{E} \cup \{(s_t, a_t, r_t)\}$
13:             **end for**
14:         **else if** $\tau \in \mathcal{D}_V$ **then**
15:             With probability $\phi$ sample batch of demonstrations
16:             **for** steps $1, 2, \ldots, T$ **do**
17:                 Replay a transition from buffer $s_t, a_t, r_t, s_{t+1} \sim \pi_{\mathcal{D}_V}(a_t|s_t)$
18:                 Store transition $\mathcal{E} \leftarrow \mathcal{E} \cup \{(s_t, a_t, r_t)\}$
19:             **end for**
20:         **else if** $\tau \in \text{Env}$ **then**
21:             With probability $1 - \rho - \phi$ collect set of trajectories by running policy $\pi_\theta$ :
22:             **for** steps $1, 2, \ldots, T$ **do**
23:                 Execute an action in the environment $s_t, a_t, r_t, s_{t+1} \sim \pi_\theta(a_t|s_t)$
24:                 Store transition $\mathcal{E} \leftarrow \mathcal{E} \cup \{(s_t, a_t, r_t)\}$
25:             **end for**
26:         **end if**
27:         Compute advantages estimates $\hat{A}_1, \hat{A}_2, \ldots, \hat{A}_T$
28:     **end for**
29:     Optimize $L^{PPO}$ wrt $\theta$, with $K$ epochs and minibatches size $M \leq NT$
30:     $\theta \leftarrow \theta - \eta \nabla_\theta L^{PPO}$
31:     Empty rollout storage $\mathcal{E} \leftarrow \{\}$
32: **end for**

---

### 3.1 SELF-IMITATION AND PRIORITIZED EXPERIENCE REPLAY

We hold the total size of the buffers $|\mathcal{D}|$ constant throughout the training. At the beginning of the training we only provide the agent with one demonstration trajectory, so $|\mathcal{D}_R| = 1$ and $|\mathcal{D}_V| = |\mathcal{D}| - 1$ as $|\mathcal{D}| = |\mathcal{D}_V| + |\mathcal{D}_R|$. $\mathcal{D}_V$ is only needed at the beginning to collect the first successful trajectories. As $|\mathcal{D}_R|$ increases and we have more variety of trajectories in $\mathcal{D}_R$, we decrease the probability $\phi$ of replaying trajectories from $\mathcal{D}_V$. After $|\mathcal{D}_R|$ is large enough, replaying the trajectories from $\mathcal{D}_V$ is more of an hindrance. The main reason for treating these two buffers separately is to slowly anneal from one to the other avoiding the hindrance in the second phase.

$|\mathcal{D}_V|$ and $|\mathcal{D}_R|$ are the quantities that are annealed according to the following formulas, each time a new successful trajectory is found and is added to $\mathcal{D}_R$ :

$$\rho = \rho + \frac{\phi_0}{|\mathcal{D}_V|_0}; \quad \phi = \phi - \frac{\phi_0}{|\mathcal{D}_V|_0}; \quad |\mathcal{D}_V| = |\mathcal{D}_V| - 1; \quad |\mathcal{D}_R| = |\mathcal{D}_R| + 1$$

where $|\mathcal{D}_R|_0$ and $|\mathcal{D}_V|_0$ are the maximum size respectively for $\mathcal{D}_R$ and $\mathcal{D}_V$ and $\phi_0$ is the value of $\phi$ at the beginning of the training. We define the probability of sampling trajectory $\tau_i$ as $P(i) = \frac{p_i^\alpha}{\sum_k p_k^\alpha}$, where $p_i = \max_t V_\theta(s_t)$, and $\alpha$ is a hyperparameter. We shift the value of $p_i$ for all the trajectories in $\mathcal{D}_V$ so as to guarantee $p_i \geq 0$. We only keep up to $|\mathcal{D}_V|$ unsuccessful trajectories at any given time.

Values are only updated for the replayed transitions. Successful trajectories are similarly sampled from $\mathcal{D}_R$ with a uniform distribution (a better strategy could be to sample according to their length), and the buffer is updated following a FIFO strategy. We introduced the value-based experience replay because we get to a level of complexity in some tasks that we could not solve by using self-imitation solely based on the reward. These are the tasks that the agent has trouble solving even once because the sequence of actions is too long or complicated. We prefer the value estimate as a selection criteria rather than the more commonly used TD-error as we speculate it is more effective at retaining promising trajectories over long periods of time. Crucially for the unsuccessful trajectories it is possible for the maximum value estimate not to be zero when the observations are similar to the ones seen in the demonstration.

We think that this plays a role in countering the effects of catastrophic forgetting, thus allowing the agent to combine separately learned sub-behaviors in a successful policy. We

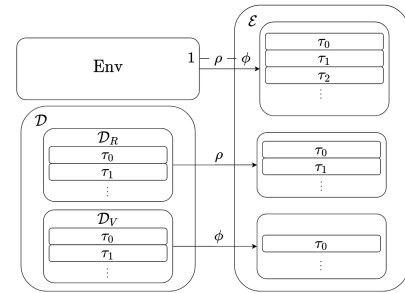

Figure 1: **Learner diagram.** The PPO+D learner samples batches that are a mixture of demonstrations and the experiences the agent collects by interacting with the environment.

illustrate this with some example trajectories of the agent shown in the Appendix in Figure 7. The value estimate is noisy and as a consequence of that, trajectories that have a low maximum value estimate on first visits may not be replayed for a long time or never as pointed out in Schaul et al. (2015). However, for our strategy to work it is enough for some of the promising trajectories to be collected and replayed by this mechanism.

## 4 EXPERIMENTS

### 4.1 THE ANIMAL-AI OLYMPICS CHALLENGE ENVIRONMENT

The recent successes of deep reinforcement learning (DRL) (Mnih et al., 2015; Silver et al., 2017; Schulman et al., 2017; Schrittwieser et al., 2019; Srinivas et al., 2020) have shown the potential of this field, but at the same time have revealed the inadequacies of using games (such as the ATARI games (Bellemare et al., 2013)) as a benchmark for intelligence. These inadequacies have motivated the design of more complex environments that will provide a better measure of intelligence.

The Obstacle Tower Environment (Juliani et al., 2019), the Animal AI Olympics (Crosby et al., 2019), the Hard-Eight Task Suite (Paine et al., 2019) and the DeepMind Lab (Beattie et al., 2016) all exhibit sparse rewards, partial observability and highly variable initial conditions. For all the tests we use the Animal-AI Olympics challenge environment. The aim of the Animal-AI Olympics is to translate animal cognition into a benchmark of cognitive AI (Crosby et al., 2019).

The environment contains an agent enclosed in a fixed size arena. Objects can spawn in this arena, including positive and negative rewards (green, yellow and red spheres) that the agent must obtain or avoid. This environment has basic physics rules and a set of objects such as food, walls, negative-reward zones, movable blocks and more. The playground can be configured by the participants and they can spawn any combination of objects in preset or random positions. We take advantage

of the great flexibility allowed by this environment to design hard-exploration problems for our experiments.

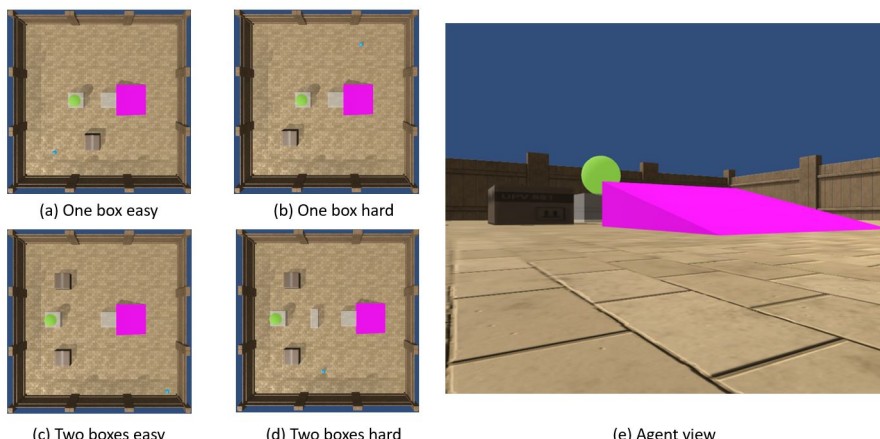

(a) One box easy      (b) One box hard

(c) Two boxes easy      (d) Two boxes hard      (e) Agent view

Figure 2: **Tasks.** In each of the tasks there is only one source of reward and the position of some of the objects is random, so each episode is different. The agent has no access to the aerial view, instead it partially observes the world through a first person view of the environment. All of the tasks are either inspired or adapted from the test set in the Animal-AI Olympics competition.

## 4.2 EXPERIMENTAL SETTING

We designed our experiments in order to answer the following questions: Can the agent learn a policy in a non-deterministic hard-exploration environment only with one human demonstration? Is the agent able to use the human demonstration to learn to solve a problem with different initial conditions (agent and boxes initial positions) than the demonstration trajectory? How does the hyperparameter $\phi$ influence the performance during training? The four tasks we used to evaluate the agent are described as follows:

- **One box easy**
  The agent has to move a single box, always spawned at the same position, in order to bridge a gap and be able to access the reward once it climbs up the ramp (visible in pink). The agent can be spawned in the range $(X : 0.5 - 39.5, Y : 0.5 - 39.5)$ if an object is not already present at the same location (Fig. 2a).
- **One box hard**
  The agent has to move a single box in order to bridge a gap and be able to access the reward, this time two boxes are spawned at any of four positions $A : (X : 10, Y : 10), B : (X : 10, Y : 30), C : (X : 30, Y : 10), D : (X : 30, Y : 30)$. The agent can be spawned in the range $(X : 0.5 - 39.5, Y : 0.5 - 39.5)$ if an object is not already present at the same location (Fig. 2b).
- **Two boxes easy**
  The agent has to move two boxes in order to bridge a larger gap and be able to access the reward, this time two boxes are spawned at any of four positions $A : (X : 10, Y : 10), B : (X : 10, Y : 30), C : (X : 30, Y : 10), D : (X : 30, Y : 30)$. The agent can be spawned in the range $(X : 15.0 - 20.0, Y : 0.5 - 15.0)$ if an object is not already present at the same location (Fig. 2c).
- **Two boxes hard**
  The agent has to move two boxes in order to bridge a larger gap and be able to access the reward. Two boxes are spawned at two fixed positions $A : (X : 10, Y : 10), B : (X : 10, Y : 30)$. A wall acts as a barrier in the middle of the gap, to prevent the agent from "surfing" a single box. The agent can be spawned in the range $(X : 15.0 - 20.0, Y : 5.0 - 10.0)$ if an object is not already present at the same location (Fig. 2d).

## 5 RESULTS

### 5.1 COMPARISON WITH BASELINES AND GENERALIZATION

In Figure 3 we compare PPO+D with parameters $\rho = 0.1, \phi = 0.3$ to the behavioral cloning baselines (with 100 and 10 demonstrations), to GAIL (with 100 demonstrations) and with PPO+BC. PPO+BC combines PPO and BC in an a similar way to PPO+D: with probability $\rho$ a sample is drawn from the demonstrations and the policy is updated using the BC loss. The value loss function remains unchanged during the BC update.

We test the GAIL implementation on a simple problem to verify it is working properly (see Section C in the Appendix). For behavioral cloning we trained for 3000 learner steps (updates of the policy) with learning rate $10^{-5}$. It is clear that PPO+D is able to outperform the baseline in all four problems. The performance of PPO+D varies considerably from task to task. This reflects the considerable difference in the range of the initial conditions for different tasks. In the tasks "One box easy" and "Two boxes hard" only the position of the agent sets different instances of the task apart. The initial positions of the boxes only play a role in the tasks "One box hard" and "Two boxes easy". Under closer inspection we could verify that in these two tasks the policy fails to generalize to configurations of boxes that are not seen in the demonstration, but does generalize well in all tasks for very different agent starting positions and orientations.

This could be because there is a smooth transition between some of the initial conditions. Due to this fact, if the agent is able to generalize even only between initial conditions that are close it will be able to progressively learn to perform well for all initial conditions starting from one demonstration. In other words the agent is automatically using a curriculum learning strategy, starting from the initial conditions that are closer to the demonstration. This approach fails when there is an abrupt transition between initial conditions, such as for different boxes configurations.

During training we realized that the policy "cheated" in the task "Two boxes easy" as it managed to "surf" one of the boxes, in this way avoiding to move the remaining box (a similar behavior was reported in Baker et al. (2019)). Although this is of itself remarkable, we were interested in testing the agent for a more complex behavior, which it avoids by inventing the "surfing" behavior. To make sure it is capable of such more complex behavior we introduced "Two boxes hard". We decided to reduce the range of the initial conditions in this last task, as we already verified that the agent can learn from such variable conditions in tasks "One box hard" and "Two boxes easy". This last experiment only tests the agent for a more complex behaviour. In the tasks "One box hard" and "Two boxes easy" the agent could achieve higher performance given more training.

We emphasise that PPO+D is designed to perform well on hard-exploration problems with stochastic environment and variable different conditions, we tested it on the of the Atari environment "BreakoutNoFrameskip-v4" and conclude that it does not lead to better performance than vanilla PPO when the reward is dense. PPO+D also learns to complete the task although more slowly than PPO.

### 5.2 THE ROLE OF $\phi$ AND THE VALUE-BUFFER $\mathcal{D}_\mathcal{V}$

In Figure 4 we mainly compare PPO+D with $\rho = 0.1, \phi = 0.3$ to $\rho = 0.5, \phi = 0.0$. Crucially, the second parameter configuration does not use the buffer $\mathcal{D}_V$. It is evident in the task "Two boxes easy" that $\mathcal{D}_V$ is essential for solving harder exploration problems. In the "One box easy" task we can see that occasionally on easier tasks not having $\mathcal{D}_V$ can result in faster learning. However, this comes with a greater variance in the training performance of the agent, as sometimes it completely fails to learn.

In Figure 4 we also run vanilla PPO ($\rho = 0.0, \phi = 0.0$) on the task "One box easy" and establish its inability to solve the task even once on any seed and any initial condition. This being the easiest of all tasks, we consider unlikely the event of vanilla PPO successfully completing any of the other harder tasks. We defer an ablation study of the parameter $\rho$ to section B in the Appendix.Although we can not make a direct comparison with Paine et al. (2019), we think it is useful to underline some of the differences between the two approaches both in the problem setting and in the performance. We attempted to recreate the same complexity of the tasks on which Paine et al. (2019) was tested. In the article, variability is introduced in the tasks on multiple dimensions such as position and

orientation of the agent, shape and color of objects, colors of floor and walls, number of objects in the environment and position of such objects. The range of the initial conditions for each of these factors was not reported. In our work we change the initial position and orientation of the agent as well as the initial positions of the boxes. As for the number of demonstrations in Paine et al. (2019) the agent has access to a hundred demonstrations, compared to only one in our work. In the present article, the training time ranges from under 5 millions frames to 500 millions, whereas in Paine et al. (2019) it ranges from 5 billions to 30. Although we adopted the use of the parameter $\rho$ from Paine et al. (2019) its value differs considerably, which we attribute to the difference between the two algorithms: one being based on PPO, the other on DQN.

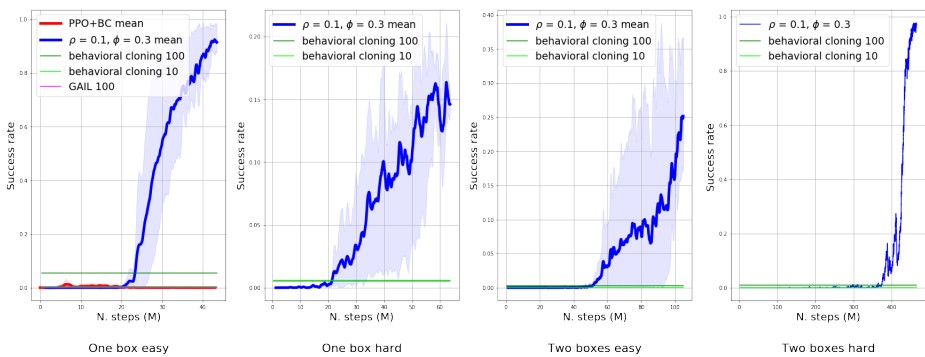

Figure 3: **Experiments.** Performance of behavioural cloning with ten and a hundred recorded human demonstrations and PPO+D with $\rho = 0.1, \phi = 0.3$ and just one demonstration. The curves represent the mean, min and max performance for each of the baselines across 3 different seeds. The BC agent sporadically obtains some rewards. GAIL with a hundred demonstrations never achieves any reward. PPO+BC has only access to one demonstration, like PPO+D. It occasionally solves the task but it is unable to archive high performance.

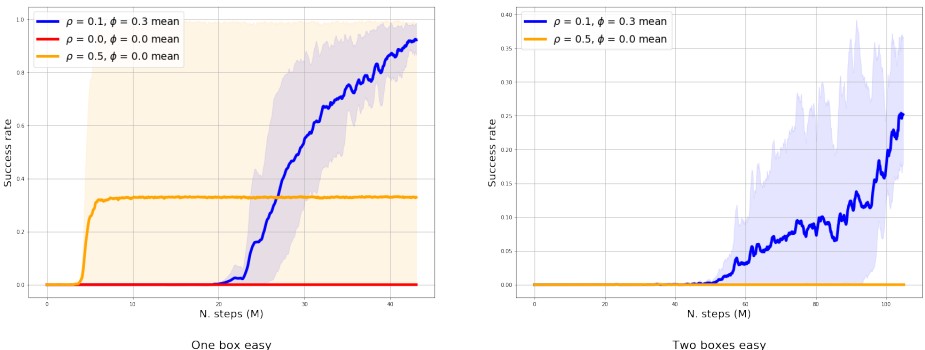

Figure 4: **Experiments.** Performance for vanilla PPO ($\rho = 0.0, \phi = 0.0$), PPO+D with $\rho = 0.5, \phi = 0.0$ and PPO+D with $\rho = 0.1, \phi = 0.3$ on the tasks "One box easy" and "Two boxes easy" using a single demonstration. Some of the curves overlap each other as they receive zero or close to zero reward. Vanilla PPO never solves the task.

## 6 CONCLUSION

We introduce PPO+D, an algorithm that uses a single demonstration to explore more efficiently in hard-exploration problems. We further improve on the algorithm by introducing two replay buffers: one containing the agent own successful trajectories as it collects these over the training and the second collecting unsuccessful trajectories with a high maximum value estimate. In the second buffer the replay of the trajectories is prioritized according to the maximum estimated value. We show that training with both these buffers solves, to some degree, all tasks the agent was presented with. We also show that vanilla PPO as well as PPO+D without the value-buffer fails to learn

the same tasks. In the article, we justify the choice of such adjustments as measures to counter catastrophic forgetting, a problem that afflicts PPO particularly. The present algorithm suffers some limitations as currently it fails to generalize to all variations of some of the problems, yet it achieves to solve several very hard exploration problems with a single demonstration. We propose to address these limitations in future work.

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

## A    TRAINING DETAILS

For the training we used 14 parallel environments and we compute the gradients using the Adam optimizer (Kingma & Ba, 2014) with fixed learning rate of $10^{-5}$. The agent perceives the environment through a 84 by 84 RGB pixels observations in a stack of 4. At each time-step the agent is allowed to take one of nine actions. We use the network architecture proposed in Kostrikov (2018) which includes a gated recurrent unit (GRU) (Cho et al., 2014) with a hidden layer of size 256. We ran the experiments on machines with 32 CPUs and 3 GPUs, model GeForce RTX 2080 Ti. The experiments where carried out with the following hyperparameters.

Table 1: Model and PPO Hyperparameters

| Parameter | Value |
|---|---|
| clip-param | 0.15 |
| gamma | 0.998 |
| frame-skip | 2 |
| frame-stack | 4 |
| num-steps | 1000 |
| num-mini-batch | 6 |
| entropy-coef | 0.02 |
| value-loss-coef | 0.1 |
| num-processes | 14 |
| lr (learning rate) | 1e-5 |
| eps (RMSprop optimizer epsilon) | 1e-5 |
| alpha (RMSprop optimizer apha) | 0.99 |
| gae-lambda | 0.95 |
| max-grad-norm | 0.5 |
| ppo-epoch | 4 |

For the training we performed no hyperparameter search over the replay ratios $\phi$ and $\rho$ but set them to a reasonable number. We found other configurations of these parameters to be sometimes more efficient in the training, such for example setting $\rho = 0.5$ and $\phi = 0.0$ in the task "One box easy". The parameters we ran all the experiments with have been chosen because they allow to solve all of the experiments with one demonstration.

In computing the probability of a trajectory to be replayed $P(i) = \frac{p_i^\alpha}{\sum_k p_k^\alpha}, \alpha = 10$. The total buffer size is $|\mathcal{D}| = 51$ with $|\mathcal{D}_V|_0 = 50$ plus the human generated trajectory. $|\mathcal{D}_R|_0 = 51$ meaning once the agent collects 50 successful trajectories, new successful trajectories overwrite old ones, following a FIFO strategy and no trajectory is replayed from the value-buffer.

The implementation used is based on the repository Kostrikov (2018). On our infrastructure we could train at approximately a speed of 1.3 millions frames per hour.

The code, pre-trained models, data-set of arenas used for training, as well as video-clips of the agent performing the tasks are available at `https://doi.org/10.6084/m9.figshare.12459602.v1`.

## B HYPERPARAMETERS ABLATION

In this section we present the results of four different experiments on a variation of the "One box easy task" where the agent position does not change across episodes and it is shared with the demonstration. We test on this variation because it is one of the simplest problems we can use to test PPO+D performance. We only perform the ablation study on $\rho$ because $\phi$ is harder to test: it is indispensable for solving difficult tasks but it can slow down the performance on easy tasks. This being an easy task, the results obtained, do not provide any insights on the effect of $\phi$ in harder problems (as shown in Figure 4 ).

The following figure shows the performance of the PPO+D algorithm where the $\rho$ parameter is changed and $\phi = 0$. Interestingly we observe that, among the values chosen, the performance peaks for $\rho = 0.3$. We hypothesize that lower $\rho$ values have worse performance because the interval between demo replays is so large that allows the network to forget the optimal policy learned with the demonstrations. On the other side, higher values of $\rho$ are even more counterproductive as they prevent the agent from learning from its own experience, most critically learning what not to do.

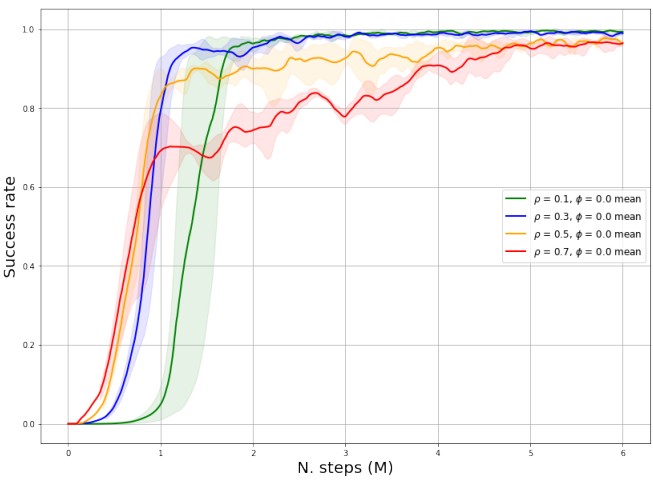

Memorize trajectory task

Figure 5: **Ablation study** Performance for PPO+D with $\rho = 0.1, \phi = 0.0$, $\rho = 0.3, \phi = 0.0$, $\rho = 0.5, \phi = 0.0$ and $\rho = 0.5, \phi = 0.0$ and PPO+D with $\rho = 0.7, \phi = 0.0$, on a variation of the "One box easy" task were the initial position of the agent is fixed. The curves represent the mean, min and max performance for each of the baselines across 3 different seeds.

## C GAIL TEST

To verify the correctness of our GAIL implementation we use for the experiments in Figure 4 we test it on a simple task in the Animal-AI environment. The task is shown in Figure 6, it consists in collecting green food of random size and position.

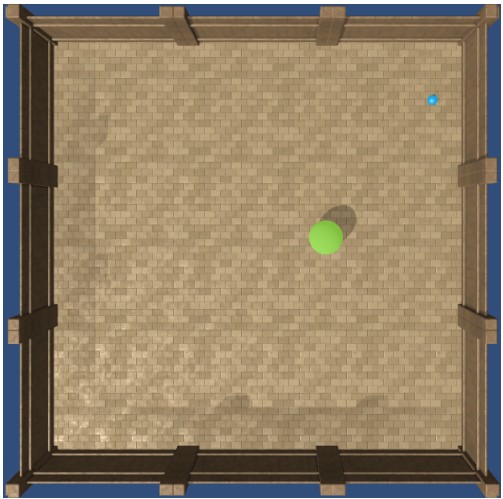

Figure 6: **Food collection task.** In this task the the agent is spawned into the arena with one green ball. The green food size and position are set randomly at the beginning of each episode. The episode ends when the green food is collected.

Our implementation of GAIL based on Li et al. (2017) was trained with the following hyperparameters besides the PPO parameters in Table 1.

Table 2: GAIL Hyperparameters

| Parameter | Value |
|---|---|
| scaling-factor | 0.001 |
| gail-epoch | 0.4 |
| gail-batch-size | 200 |

Table 3: Performance on the "Food collection task"

| Method | Avg. Success rate | Std. |
|---|---|---|
| GAIL | 0.997 | 0.045 |
| BC | 0.617 | 0.487 |

In Table 3 we report the performance of both GAIL and behaviour cloning on the simple task. We note that although both methods achieve reasonable performance, the agent trained with GAIL reaches near-perfect performance, whereas the BC agent performance tends to fluctuate significantly. The GAIL policy was trained for 120 millions time-frames and behavioral cloning we trained for 3000 learner steps (updates of the policy) with learning rate $10^{-5}$.

# D    ANALYSIS OF THE EFFECT OF THE VALUE-BUFFER

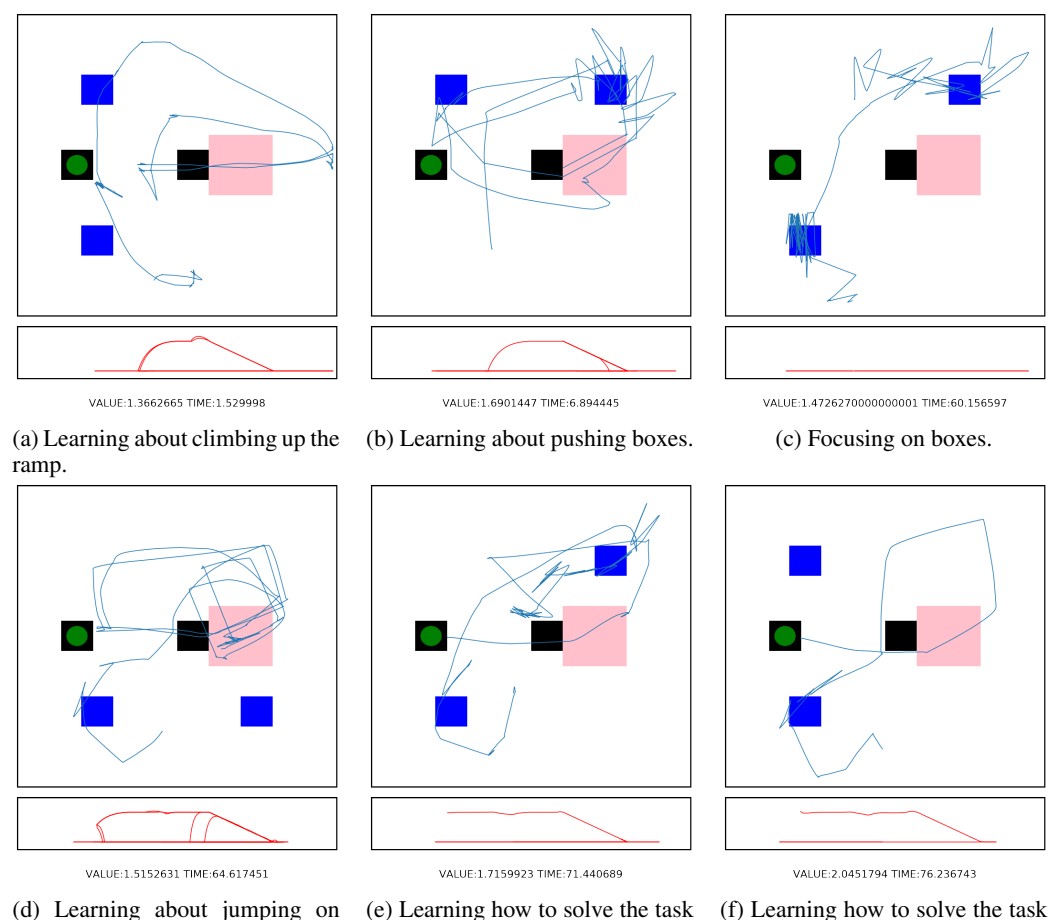

(a) Learning about climbing up the ramp.

(b) Learning about pushing boxes.

(c) Focusing on boxes.

(d) Learning about jumping on boxes.

(e) Learning how to solve the task pushing two boxes.

(f) Learning how to solve the task with only one box.

Figure 7: **Sub-behaviors.** Trajectories the agent played on the task "Two boxes easy". In each of the figure the upper part shows the movements of the agent on the X-Y plane while the lower part shows the movement on the X-Z plane. The images are ordered by the time they were executed in the training in millions of frames.

The value-buffer experience replay creates an incremental curriculum for the agent to learn, keeping different trajectories that achieved an high maximum value in the buffer incentives the agent to combine these different sub-behaviours e.g. pushing the blocks and going up the ramp.

