# OpenReview forum: "Guided Exploration with Proximal Policy Optimization using a Single Demonstration"
_ICLR.cc/2021/Conference — Reject_

### Official Review · AnonReviewer3 · 2020-10-27
**Progress on a difficult class of RL problem**

**Rating:** 6
**Confidence:** 3

**Review:**

## Summary
This paper looks at problems that have sparse rewards, are partially observable, and have
variable initial conditions.  Previous work [R2D3, Paine et al 2019] tackles this problem using
off-policy recurrent Q-learning. Instead, this work proposes to use a PPO+D, an on-policy method (PPO)
for a policy with memory (GRUs). This relies on storing successful trajectories (demonstration(s)
and discovered), and replaying these as experiences using importance sampling. Other details
such as a value-prioritized buffer, annealed use of that buffer, an entropy term, and more
are given.  The method is tested on four environment variations, as constructed in the Animal-AI Olympics challenge env.
Exact comparisons with R2D3 are not possible due to missing initial condition ranges.
It is compared to a behavior cloning baseline, vanilla PPO, and a value-buffer ablation,
It is shown that it can succeed with just one demonstration, although it fails to generalize
to all many of the variations for the problems.

## Strengths
- interesting problem setting (although this is not novel, e.g., R2D3)
- interesting exploration beyond R2D3, i.e., to on-policy methods
- use of value buffer as a means to allow for progress on harder problems

## Weaknesses
- could have compared against an implementation of some facsimile of R2D3
- lack of generalization to wider initial conditions means that the results may not be as interesting as they first seem
- behavior cloning is, unsurprisingly, a poor baseline for a problem with variable initial conditions and with partial observability
- the main result is really: "sometimes it works!".  That is actually of interest, although really a "proof by example" result.
- only tested on limited tasks, although similar to R2D3.  Thus it is unclear how generalizable the method and results will be;  it may overfit in some respects to the task at hand.

## Recommendation
Overall, while the results come with some real caveats, the paper may inspire further work in the use of demonstrations in sparse-reward settings, in conjunction with on-policy methods such as PPO. The use of the value-prioritized buffer is also a feature
that could see further adoption.

## Questions
- why could behavior cloning realistically be expected to work as a baseline, given the variable initial states?

## Additional Feedback
- DR and DV: it's unclear what the 'R' stands for in DR.  It's always helpful to define the origins of the chosen nomenclature for cases where it is not obvious.
- please tell the reader where the value estimates come from for the value buffer for unsuccessful trajectories.  Why aren't these all simply zero?
- Figure 1 is really helpful;  place it earlier in section 3?

---

> ### Author Response · Authors · 2020-11-23
> **Response**
>
> >Thanks for your positive  feedback.
>
> Weaknesses
> could have compared against an implementation of some facsimile of R2D3
>
> >We initially attempted to have a baseline with R2D3, but the  initial paper did not provide any code and the environment was also not public. The only public implementation we found is incomplete as it has missing proprietary code which was not publicly released and it was necessary for running parallel environments.
>
> lack of generalization to wider initial conditions means that the results may not be as interesting as they first seem.
>
> >We decided to frame the results around the quite remarkable fact that the agent is able to solve these very hard tasks given a *single* demonstration, in an environment with variable initial conditions. If the agent was given more demonstrations with very different initial conditions it is reasonable to believe (although should be investigated) that it would learn to solve any box configuration, when given the trajectory of a few demonstrations. With a single demonstration it is surprising the agent generalizes to different starting positions. It does not generalize to different boxes configuration. Nevertheless we added the following to section 5.1,on our guess to why PPO+D generalizes to some initial conditions and not others:
> “This could be because there is a smooth transition between some of the initial conditions. Due to this fact, if the agent is able to generalize even only between initial conditions that are close it will be able to progressively learn to perform well for all initial conditions starting from one demonstration. In other words the agent is automatically using a curriculum learning strategy, starting from the initial conditions that are closer to the demonstration. This approach fails when there is an abrupt transition between initial conditions, such as for different boxes configurations.”
>
> behavior cloning is, unsurprisingly, a poor baseline for a problem with variable initial conditions and with partial observability.
>
> >The baseline algorithm is given a hundred times more data than the PPO+D algorithm, each one with a different initial condition, whereas PPO+D only repeats the same trajectory with the same initial condition. Nevertheless, we have now also added more baselines such as GAIL and the PPO+BC obtaining similar results as before, i.e. PPO+D provides the best performance.
>
> the main result is really: "sometimes it works!". That is actually of interest, although really a "proof by example" result.
>
> >Our results are strictly empirical, apart from how  to insert in a mathematically correct way demonstrations into PPO.
>
> only tested on limited tasks, although similar to R2D3. Thus it is unclear how generalizable the method and results will be; it may overfit in some respects to the task at hand.
>
> > This is not a general RL algorithm nor it claims to be, and we do not expect it to perform particularly well in normal dense reward tasks. On such tasks PPO+D is likely to perform rather poorly as replaying the same trajectories is likely to be more of an hindrance than  booster of performance in this case. However, we do believe the algorithm can generalize to problems that have similar formulations, in terms of the sparse reward assignment, but are very different in the nature of the task.
>
> why could behavior cloning realistically be expected to work as a baseline, given the variable initial states?
>
> >Because BC is given a hundred times more data than the PPO+D algorithm, each one with a different initial condition, whereas PPO+D only repeats the same trajectory with the same initial condition. We also added more baselines such as GAIL and the PPO+BC suggested by reviewer 2 with similar outcomes.
>
> Additional Feedback
> DR and DV: it's unclear what the 'R' stands for in DR. It's always helpful to define the origins of the chosen nomenclature for cases where it is not obvious.
>
> >We have added a clarification of the origin of the terms as we introduce them see “... are the trajectories collected prioritizing the value estimate (V stands for value), and D_{R} (R stands for reward) contains the initial human...”
>
> please tell the reader where the value estimates come from for the value buffer for unsuccessful trajectories. Why aren't these all simply zero?
>
> >It is because the critic generalizes to yet unseen images. For instance if the agent sees something really similar to an image in the demonstration trajectory it is expected to assign it a non zero value.
> In section 3.1 we added “Crucially for the unsuccessful trajectories it is possible for the maximum value estimate not to be zero when the observations are similar to the ones seen in the demonstration.”
>
> Figure 1 is really helpful; place it earlier in section 3?
>
> >Review N.1 likes it, so we have decided to keep it, but unfortunately had to reduce its size to make room for some of the changes

---

### Official Review · AnonReviewer2 · 2020-10-28
**Baselines are very weak**

**Rating:** 4
**Confidence:** 5

**Review:**

##########################################################################

Summary:

The paper proposes a modification of the PPO algorithm which can accommodate a single task demonstration with the goal of faster learning in sparse-reward tasks. There are 4 tasks proposed by the paper, inspired by Animal-AI Olympics. The baselines are Behavioural Cloning (BC) and vanilla PPO. The proposed method outperforms those baselines.


##########################################################################

Reasons for score:

From the text it seems like BC baseline doesn't have access to rewards. If that is the case, the baselines are very weak - they either have access to demonstrations only or sparse rewards only.

The most basic fair baseline would be doing BC gradient updates to policy network from time to time during PPO training - should be a reasonable amount of work to implement. Ideally, comparison to relevant prior work should be at least attempted: Paine et al (R2D3) or some other whichever is easier to implement (see below for related work comments).


##########################################################################

Pros:

1. Nice research direction: combining demonstrations with sparse rewards.

##########################################################################

Cons:

1. More baselines are needed to evaluate the value of the proposed method. Prior work does exist (e.g., Paine et al.) and warrants more extensive comparisons.
2. Related work needs to mention some relevant papers:
- "One-Shot Imitation Learning" https://arxiv.org/pdf/1703.07326.pdf
- "Watch, try, learn: meta-learning from demonstrations and rewards" https://arxiv.org/pdf/1906.03352.pdf
3. There are some established benchmarks for sparse-reward tasks - it might be more productive to attack those as they already have some baselines. Introducing new tasks requires some discussion on why the established tasks are unsuitable for the goals of the paper. For example:
- Vizdoom navigation by Pathak et al https://arxiv.org/pdf/1705.05363.pdf
- https://openai.com/blog/ingredients-for-robotics-research/
- https://aihabitat.org/

##########################################################################

Questions during rebuttal period:

Please address and clarify the cons above.

---

> ### Author Response · Authors · 2020-11-23
> **Response**
>
> From the text it seems like BC baseline doesn't have access to rewards. If that is the case, the baselines are very weak - they either have access to demonstrations only or sparse rewards only.
>
> >BC does not have access to rewards, but there are no rewards apart from successfully completing the problem. Furthermore, BC is given a hundred times more human demonstrations than PPO+D, each one with a different initial condition. PPO+D only has  one trajectory with a single initial condition.  To improve the benchmarking, we also added more baselines such as GAIL and the PPO+BC as  suggested.
>
> The most basic fair baseline would be doing BC gradient updates to policy network from time to time during PPO training - should be a reasonable amount of work to implement. Ideally, comparison to relevant prior work should be at least attempted: Paine et al (R2D3) or some other whichever is easier to implement (see below for related work comments).
>
> >We have included the algorithm suggested, i.e. doing BC gradient updates to policy network from time to time during PPO training) in the same conditions as PPO+D with a single trajectory on the easiest task “One box easy”. The results indicated that this too performs worse than PPO+D. We have added a baseline using GAIL which also performs worse than PPO+D .  As mentioned in the paper we would have been happy to compare with R2D3 but no multi-threaded implementation of the algorithm was made available  and the environment on which they conducted the experiments was also not publicly available.
>
>
> Pros:
> Nice research direction: combining demonstrations with sparse rewards.
>
> >Thanks for your positive  feedback.
>
> Cons:
> More baselines are needed to evaluate the value of the proposed method. Prior work does exist (e.g., Paine et al.) and warrants more extensive comparisons.
>
> >As indicated above, we have included the algorithm BC+PPO and GAIL to our baselines, still we can maintain that PPO+D performs better in the current settings.
>
> Related work needs to mention some relevant papers:
> "One-Shot Imitation Learning" https://arxiv.org/pdf/1703.07326.pdf
> "Watch, try, learn: meta-learning from demonstrations and rewards" https://arxiv.org/pdf/1906.03352.pdf
>
> >We agree the two papers you mentioned are very relevant to the work presented here and have included the following in section 2: “Other approaches, such as Duan et al. (2017); Zhou et al. (2019) pursue a meta-learning strategy where the agent learns to learn from demonstrations, such approaches are perhaps the most promising, but they require at least a demonstration for each task for a subset of all tasks.”
>
> There are some established benchmarks for sparse-reward tasks - it might be more productive to attack those as they already have some baselines. Introducing new tasks requires some discussion on why the established tasks are unsuitable for the goals of the paper. For example:
> Vizdoom navigation by Pathak et al https://arxiv.org/pdf/1705.05363.pdf
> https://openai.com/blog/ingredients-for-robotics-research/
> https://aihabitat.org/
>
> >Like us Paine et al (R2D3) deem existing environments unsuitable as they do not exhibit all or some of the following characteristics: sparse rewards. partial observability and variable initial conditions. We mention this at the beginning of section 4.1, 1 and  2  as we discuss that some of the best performing algorithms on these baselines rely on the absence of  the characteristics  we list above.  We would have tested it on the environment proposed in R2D3 as we agree it fulfills these requirements, but it was not made available. As we mention in the paper we draw inspiration from that environment and try to create tasks of comparable complexity.

---

### Official Review · AnonReviewer1 · 2020-10-29
**Great work, but missing experiments or more depth in analysis**

**Rating:** 6
**Confidence:** 4

**Review:**

## Summary

The authors introduce a simple extension of PPO that uses a single expert demonstration and a modified sampling algorithm to show better performance than vanilla PPO in a difficult 3D setting.

## Strengths & Weaknesses

#### Strengths

- The paper is well-written and the method is simple, yet powerful. Learning something from a single demonstration is no easy feat.
- The videos shown in the submission look impressive in how difficult the setup is and how the agent manages to learn complex strategies like.
- Overall, the paper is very "short & sweet" in that it's not a groundbreaking new technique, but a small change to PPO but it's well explained, and the results that _are_ in the paper are good.

#### Weaknesses

- The main problem I have with this is actually the lack of further experiments. For such an easy extension of PPO, I would've expected you to have no problem running this on an Atari environment and at least one MuJoCo environment too (where it's also easy to gather a human demonstration, like controlling the reacher via inverse-kinematics or the tricky Pusher). Compared to vanilla PPO we should see improvements across the board, no?
- Similarly, you just arrived at the hyperparameters $p = 0.1, \phi=0.3$ without explanation or ablation. Do you maybe want to justify how these parameters came to be and what happens if either parameter is higher or lower?

**TL;DR how to make me raise my score:** Include at least one Atari and one Mujoco/Robot environment (since Ilya Kostrikov's implementation that you use supports these out of the box) and either add an ablation study on a single env over p/phi or explain the importance of these values.

## Impact & Recommendation

This is fundamentally good material. It's not groundbreakingly new but I think it could make for an easy-to-use imitation learning baseline that would help in a lot of scenarios get the method off the ground. However, the current paper doesn't show the rigour and depth of analysis that I would expect from an ICLR paper. I hope the authors can make up for that in the rebuttal week and then I'm happy to up my score. Currently, my recommendation is borderline. If the present method was really an all-around improvement over PPO, why did the authors not show it on a tried-and-true OpenAI Gym task but only in their own made-up setting?

## Minor Nitpicks

- I'd report a few more seeds - I think 5 seeds is a good starting point.
- Your plotting of runs is uncommon - usually, you either plot the mean and standard deviation or the mean and min/max.
- In Fig. 3 and 4, the fonts in the legend need to be bigger
- On page 6 you're twice in a row weirdly enthusiastic for Unity-ML / the "flexibility allowed by this environment". These are odd things to say in a research paper unless you're an employee of Unity
- Algorithm 1 is a bit verbose but great. Makes it very clear. On the flip side of that, Figure 1 is a bit redundant. These two things communicate the same idea and I like Algo 1 better.
- There are a few missing commas, like "In our approach,", bottom of the first page.

---

> ### Author Response · Authors · 2020-11-23
> **Response**
>
>
> >Thanks for your positive  feedback.
>
> Weaknesses
> The main problem I have with this is actually the lack of further experiments. For such an easy extension of PPO, I would've expected you to have no problem running this on an Atari environment and at least one MuJoCo environment too (where it's also easy to gather a human demonstration, like controlling the reacher via inverse-kinematics or the tricky Pusher). Compared to vanilla PPO we should see improvements across the board, no?
>
> >Like us, Paine et al. (R2D3) deem existing environments unsuitable as they do not exhibit all or some of the following characteristics: sparse rewards, partial observability and variable initial conditions. We expect PPO+D to perform well in environments that have these characteristics.  For instance the fact that we replay certain trajectories to counter the effect of catastrophic forgetting, if used in tasks where catastrophic forgetting is not a problem to begin with, will slow down the training. We tried running PPO+D on the BreakoutNoFrameskip-v4 of the Atari environment and we concluded PPO+D leads to worse performance than vanilla PPO, although it also learns to complete the task, it is considerably slower.
> We have added to the manuscript: “We emphasise that PPO+D is designed to perform well on hard-exploration problems with stochastic environment and variable different conditions, we tested it on the  of the Atari environment “BreakoutNoFrameskip-v4” and conclude that it does not lead to better performance than vanilla PPO when the reward is dense. PPO+D also learns to complete the task although more slowly than PPO” at the end of section 5.1.
>
> Similarly, you just arrived at the hyperparameters �=0.1,�=0.3 without explanation or ablation. Do you maybe want to justify how these parameters came to be and what happens if either parameter is higher or lower?
>
> >We chose the value of these hyperparameters by trial and error. We added now an ablation study in the Appendix.
>
> how to make me raise my score: Include at least one Atari and one Mujoco/Robot environment (since Ilya Kostrikov's implementation that you use supports these out of the box) and either add an ablation study on a single env over p/phi or explain the importance of these values.
>
> >We tried running PPO+D on the BreakoutNoFrameskip-v4 of the Atari environment and we concluded PPO+D leads to worse performance than vanilla PPO, although it also learns to complete the task, it is considerably slower.
> We have added to the manuscript: “We emphasise that PPO+D is designed to perform well on hard-exploration problems with stochastic environment and variable different conditions, we tested it on the  of the Atari environment “BreakoutNoFrameskip-v4” and conclude that it does not lead to better performance than vanilla PPO when the reward is dense. PPO+D also learns to complete the task although more slowly than PPO” at the end of section 5.1.
> We added now an ablation study in the Appendix.
>
> Impact & Recommendation
> ... If the present method was really an all-around improvement over PPO, why did the authors not show it on a tried-and-true OpenAI Gym task but only in their own made-up setting?
>
> >See answer right above
>
> Minor Nitpicks
> I'd report a few more seeds - I think 5 seeds is a good starting point.
>
> >We tried but finally we could not run five seeds for all of the experiments due to lack of resources. We  hope  3 seeds are acceptable.
>
> Your plotting of runs is uncommon - usually, you either plot the mean and standard deviation or the mean and min/max.
>
> >We have now plotted mean and min/max as suggested.
>
> In Fig. 3 and 4, the fonts in the legend need to be bigger
>
> >Fixed.
>
> On page 6 you're twice in a row weirdly enthusiastic for Unity-ML / the "flexibility allowed by this environment". These are odd things to say in a research paper unless you're an employee of Unity
>
> >We have removed the sentence about Unity.  Although in this sentence: “We take advantage of the great flexibility allowed by this environment to design hard-exploration problems for our experiments” we refer to the Animal AI environment and not ML-Agents.
>
> Algorithm 1 is a bit verbose but great. Makes it very clear. On the flip side of that, Figure 1 is a bit redundant. These two things communicate the same idea and I like Algo 1 better.
>
> >We have kept Figure 1 as other reviewers specifically find it helpful, we have reduced its size to make space for some of the changes.
>
> There are a few missing commas, like "In our approach,", bottom of the first page.
> >We have revised the text for this type of error.

---

> > ### Comment · AnonReviewer1 · 2020-11-24
> > **Good changes, keep going**
> >
> > Dear authors,
> >
> > Thanks for the changes, clarifications, and comments.
> >
> > I'll not yet update my rating because you didn't address my comment about MuJoCo environments. The ones that are built into OpenAI Gym fulfill your criteria (random init and partial observability are given; sparse reward is trivial to get from the dense reward).
> >
> > And how long are your experiments that you don't have resources for 5 seeds? Usually, when I run PPO for 1mil steps with a single CPU and a single GPU, it takes 1h-2.5h, depending on the environment.

---

> > > ### Author Response · Authors · 2020-11-24
> > > **Further explanation**
> > >
> > > Thanks for the changes, clarifications, and comments.
> > >
> > > >Thank you
> > >
> > > I'll not yet update my rating because you didn't address my comment about MuJoCo environments. The ones that are built into OpenAI Gym fulfill your criteria (random init and partial observability are given; sparse reward is trivial to get from the dense reward).
> > >
> > > > We did initially try the Halfcheetah environment even if this environment is not partially observable. We could not get PPO+D to work on it and we did not feel confident enough if it was due to a bug in the adaptation of PPO+D for this environment, the fact that the continuous action space might need some adjustments to the algorithm or something else. For instance the fact that mujoco terminates the episode when the agent falls close to the ground affects the proportion of environment experience/demo replay since the way we regulate this in the algorithm relies on the episodes and not on the frames directly. This meant that most of the frames were repeating the demonstration for PPO+D, as the normal policy would terminate the episode almost immediately.
> > > The experiments we tried with halfcheetah were with a sparsified reward. The fact that such a task is solvable when the reward is given only once the agent covers a certain distance, is not clear at all: the original reward function punishes large movements besides rewarding the distance and it is hard to imagine that such a reward would have the same effect when not given at each timestep. The fact we have so many doubts on this task made us omit this result, because there is really nothing we can say with confidence.
> > > We switched to Atari because due to the discrete action space they are more similar to the Animal-AI environment and we discussed the results on Atari, where we could make PPO+D work, even though due to the dense rewards, it is not ideally suited for it. .
> > >
> > > And how long are your experiments that you don't have resources for 5 seeds? Usually, when I run PPO for 1mil steps with a single CPU and a single GPU, it takes 1h-2.5h, depending on the environment.
> > >
> > > > Doing the extra 2 seeds would have meant 100*2 + 60*2 +40*2 millions for the first plot and 40*2 + 40*2 + 100*2 for the second plot. Since the other reviewer requested more baselines, that would have meant for these we world have needed: GAIL: 40*5 + 120*5 PPO+BC: 5*40 and 6*5*4 for the ablation study.
> > > These sum up to 1880 million frames to run on the 6x8 cores CPUs and 12 GPUs that we had available, it would have been quite difficult to make it in time. At the frame rate of AnimalAI, which is a slow environment due to the 3D rendering engine, we could perform around 300 frames per second on 16 cores+1GPU, meaning more than 20 days to finish. We thus made a choice to focus on porting PPO+D to work on Atari/Mujoco and adding the baselines requested by the other reviewers, rather than adding seeds as the difference between PPO+D and the other baselines is substantial and statistical significant already with 3 seeds (we use min/max in the plots not +-std).

---

### Decision · Program_Chairs · 2021-01-07
**Final Decision**

**Decision:**

Reject

**Comment:**

There was quite a bit of internal discussion on this paper. To summarize:
- The idea is very neat and interesting and likely to work
- The paper is likely to inspire future work
- There are still serious doubts  about the experimental evaluation that is not entirely up to par with current standards
  - The reviewers were not convinced 100% by the arguments about the 'custom' environments
  - The reviewers were not convinced 100% that the baselines were given their best shot

While the paper has potential to provide valuable input for the community, it needs a bit more work before being presentable at a highly competitive venue like ICLR.